# Learning What Not to Model: Gaussian Process Regression with Negative Constraints

## Abstract

We empirically demonstrate that our $\mathcal{GP}$-NC framework performs better than the traditional $\mathcal{GP}$ learning and that our framework does not affect the scalability of Gaussian Process regression and helps the model converge faster as the size of the data increases. Gaussian Process ($\mathcal{GP}$) regression fits a curve on a set of datapairs, with each pair consisting of an input point 'x' and its corresponding target regression value '$y(\mathbf{x})$' (a *positive* datapair). But, what if for an input point '$\bar{\mathbf{x}}$', we want to constrain the $\mathcal{GP}$ to avoid a target regression value '$\bar{y}(\bar{\mathbf{x}})$' (a *negative* datapair)? This requirement can often appear in real-world navigation tasks, where an agent would want to avoid obstacles, like furniture items in a room when planning a trajectory to navigate. In this work, we propose to incorporate such negative constraints in a $\mathcal{GP}$ regression framework. Our approach, '$\mathcal{GP}$-NC' or Gaussian Process with Negative Constraints, fits over the positive datapairs while avoiding the negative datapairs. Specifically, our key idea is to model the negative datapairs using small blobs of Gaussian distribution and maximize its KL divergence from the $\mathcal{GP}$. We jointly optimize the $\mathcal{GP}$-NC for both the positive and negative datapairs. We empirically demonstrate that our $\mathcal{GP}$-NC framework performs better than the traditional $\mathcal{GP}$ learning and that our framework does not affect the scalability of Gaussian Process regression and helps the model converge faster as the size of the data increases.

## 1 Introduction

Gaussian process are one of the most studied model class for data-driven learning as these are nonparametric, flexible function class that requires little prior knowledge of the process. Traditionally, GPs have found their applications in various fields of research, including Navigation systems (*e.g.*, in Wiener and Kalman filters) (Jazwinski, 2007), Geostatistics, Meteorology (Kriging (Handcock & Stein, 1993)) and Machine learning (Rasmussen, 2006). The wide range of applications can be attributed to the property of GPs to model the target uncertainty by providing the predictive variance over the target variable.

Gaussian process regression in its current construct fits only on a set of *positive* datapairs, with each pair consisting of an input point and its desired target regression value, to learn the distribution on a functional space. However, in some cases, more information is available in the form of datapairs, where at a particular input point, we want to avoid a range of regression values during the curve fitting of $\mathcal{GP}$. We designate such data as *negative* datapairs.

An illustration where modeling such negative datapairs would be extremely beneficial is given in Fig 1. In Fig 1(b), an agent wants to model a trajectory such that it covers all the positive datapairs marked by 'x'. However, it is essential to note that the agent would run into an obstacle if it models its trajectory based only on the positive datapairs. We can handle this problem of navigating in the presence of obstacles in two ways, one way is to get a high density of positive datapairs near the obstacle, and the other more straightforward approach is to just mark the obstacle as a negative datapair. The former approach would unnecessarily increase the number of positive datapairs for $\mathcal{GP}$ to regress. Hence, it may run into scalability issues. However, in the latter approach, if the point is denoted as a negative datapair with a sphere of negative influence around it as illustrated by Fig 1.c, the new trajectory can be modeled with less number of datapairs that accounts for all obstacles on the

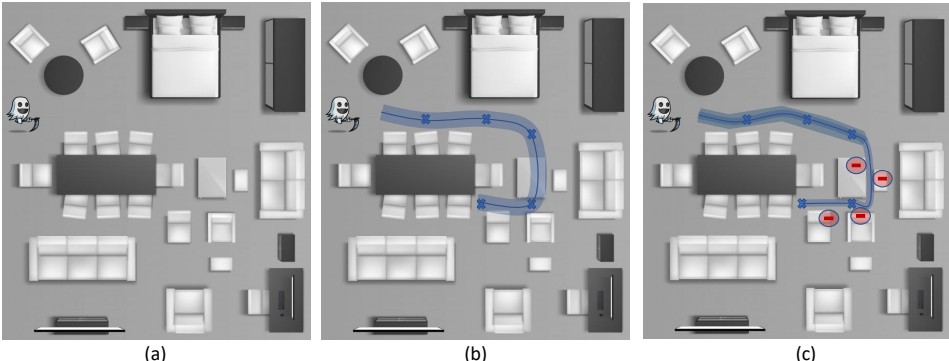

(a)                              (b)                              (c)

Figure 1: An illustration of our problem setup. (a) top view of the room where the agent wants to travel to a particular location while avoiding obstacles; (b) the agent has been given the location of the positive datapairs that are needed to be covered in its trajectory. Since the number of these observed points is low, the agent is not able to avoid the obstruction (coffee table) while forecasting its course; (c) the agent is given both the positive datapairs which it needs to reach along with negative datapairs (area of influence is given by shaded red region) that should be avoided during the modeling of future trajectory.

way. Various $\mathcal{GP}$ methods in their current framework lack the ability to incorporate these negative datapairs for the regression paradigm.

**Contributions**:In this paper, we explore the concept of negative datapairs. We provide a simple yet effective $\mathcal{GP}$ regression framework, called $\mathcal{GP}$-NC which can fit on the positive datapairs while avoiding the negative datapairs. Specifically, our key idea is to model the negative datapairs using a small Gaussian blob and maximize its KL divergence from the $\mathcal{GP}$. Our framework can be easily incorporated for various types of $\mathcal{GP}$ models (*e.g.*, exact, SVGP (Hensman et al., 2013), PPGPR (Jankowiak et al., 2019)) and works well in the scalable settings too. We empirically show in §5 that the inclusion of negative datapairs in training helps with both the increase in accuracy and the convergence rate of the algorithm.

## 2 REVIEW OF GAUSSIAN PROCESS REGRESSION

We briefly review the basics of Gaussian Process regression, following the notations in (Wilson et al., 2015). For more comprehensive discussion of $\mathcal{GP}$s, refer to (Rasmussen, 2006).

A Gaussian process is a collection of random variables, any finite number of which have a joint Gaussian distribution (Rasmussen, 2006). We consider a dataset $\mathcal{D}$ with $n$ D-dimensional input vectors, $X = \{\mathbf{x}_1, \cdots, \mathbf{x}_n\}$ and corresponding $n \times 1$ vector of targets $\mathbf{y} = (y(\mathbf{x}_1), \cdots, y(\mathbf{x}_n))^T$. The goal of $\mathcal{GP}$ regression is to learn a function $f$ that maps elements from input space to a target space, *i.e.*, $y(\mathbf{x}) = f(\mathbf{x}) + \epsilon$ where $\epsilon$ is i.i.d. noise. If $f(\mathbf{x}) \sim \mathcal{GP}(\mu, k_\theta)$, then any collection of function values $\mathbf{f}$ has a joint multivariate normal distribution given by,

$$\mathbf{f} = f(X) = [f(\mathbf{x}_1), \cdots, f(\mathbf{x}_n)]^T \sim \mathcal{N}(\mu_X, K_{X,X}) \tag{1}$$

with the mean vector and covariance matrix defined by the functions of the Gaussian Process, as $(\mu_X)_i = \mu(x_i)$ and $(K_{X,X})_{ij} = k_\theta(\mathbf{x}_i, \mathbf{x}_j)$. The kernel function $k_\theta$ of the $\mathcal{GP}$ is parameterized by $\theta$. Assuming additive Gaussian noise, $y(\mathbf{x})|f(\mathbf{x}) \sim \mathcal{N}(y(\mathbf{x}); f(\mathbf{x}), \sigma^2)$, then the predictive distribution of the $\mathcal{GP}$ evaluated at the $n_*$ test points indexed by $X_*$, is given by

$$\begin{aligned} \mathbf{f}_* | X_*, X, \mathbf{y}, \theta, \sigma^2 &\sim \mathcal{N}(E[\mathbf{f}_*], \text{cov}(\mathbf{f}_*)), \\ E[\mathbf{f}_*] &= \mu_{X_*} + K_{X_*,X}[K_{X,X} + \sigma^2 I]^{-1}\mathbf{y}, \\ \text{cov}(\mathbf{f}_*) &= K_{X_*,X_*} - K_{X_*,X}[K_{X,X} + \sigma^2 I]^{-1}K_{X,X_*} \end{aligned} \tag{2}$$

$K_{X_*,X}$ represents the $n_* \times n$ covariance matrix between the $\mathcal{GP}$ evaluated at $X_*$ and $X$. Other covariance matrices follow similar conventions. $\mu_{X_*}$ is the mean vector of size $n_* \times 1$ for the test points and $K_{X,X}$ is the $n \times n$ covariance matrix calculated using the training inputs $X$. The underlying hyperparameter $\theta$ implicitly affects all the covariance matrices under consideration.

## 2.1 $\mathcal{GP}$S: LEARNING AND MODEL SELECTION

We can view the $\mathcal{GP}$ in terms of fitting a joint probability distribution as,

$$p\left(\mathbf{y}, \mathbf{f} | X\right) = p\left(\mathbf{y} | \mathbf{f}, \sigma^2\right) p\left(\mathbf{f} | X\right) \tag{3}$$

and we can derive the marginal likelihood of the targets $\mathbf{y}$ as a function of kernel parameters alone for the $\mathcal{GP}$ by integrating out the functions $\mathbf{f}$ in the joint distribution of Eq. (3). A nice property of the $\mathcal{GP}$ is that this marginal likelihood has an analytical form given by,

$$\mathcal{L}(\theta) = \log p(\mathbf{y}|\theta, X) = -\frac{1}{2}\left(\mathbf{y}^T \left(K_\theta + \sigma^2 I\right)^{-1} \mathbf{y} + \log\left(\left|K_\theta + \sigma^2 I\right|\right) + N \log\left(2\pi\right)\right) \tag{4}$$

where we have used $K_\theta$ as a shorthand for $K_{X,X}$ given $\theta$. The process of kernel learning is that of optimizing Eq. (4) w.r.t. $\theta$.

The first term on the right hand side in Eq. (4) is used for model fitting, while the second term is a complexity penalty term that maintains the Occam's razor for realizable functions as shown by (Rasmussen & Ghahramani, 2001). The marginal likelihood involves matrix inversion and evaluating a determinant for $n \times n$ matrix, which the naive implementation would require a cubic order of computations $\mathcal{O}(n^3)$ and $\mathcal{O}(n^2)$ of storage. Approaches like Scalable Variational GP (SVGP) (Hensman et al., 2013) and parametric GPR (PPGPR) (Jankowiak et al., 2019) have proposed approximations that lead to much better scalability. Please refer to Appexdix A for details.

## 3 $\mathcal{GP}$ REGRESSION WITH NEGATIVE DATAPAIRS

As shown in Fig. 1, we want the model to avoid certain negative datapairs in its trajectory. In other words, we want the trajectory of the Gaussian Process to have a very low probability of passing through these negative datapairs. In this section, we will first formalize the functional form of the negative datapairs and then subsequently describe our framework called $\mathcal{GP}$-NC regression.

### 3.1 DEFINITION OF POSITIVE & NEGATIVE DATAPAIRS

*Positive datapairs*: The set of datapairs through which the $\mathcal{GP}$ should pass are defined as positive datapairs. We assume a set of $n$ datapairs (input, positive target) with D-dimensional input vectors, $X = \{\mathbf{x}_1, \cdots, \mathbf{x}_n\}$ and corresponding $n \times 1$ vector of target regression values $\mathbf{y} = \{y(\mathbf{x}_1), \cdots, y(\mathbf{x}_n)\}$.

*Negative datapairs*: The set of datapairs which the $\mathcal{GP}$ should avoid (obstacles) are defined as negative datapairs. We assume a set of $m$ datapairs (input, negative target) with D-dimensional input vectors $\bar{X} = \{\bar{\mathbf{x}}_1, \cdots, \bar{\mathbf{x}}_m\}$ and corresponding set of negative targets $\bar{\mathbf{y}} = \{\bar{y}(\bar{\mathbf{x}}_1), \cdots, \bar{y}(\bar{\mathbf{x}}_m)\}$. The sample value of $\mathcal{GP}$ at input $\bar{\mathbf{x}}_i$, given by $f(\bar{\mathbf{x}}_i)$, should be far from the negative target regression value $\bar{y}(\bar{\mathbf{x}}_i)$.

Note that it is possible that a particular input $\mathbf{x}$ can be in both the positive and negative data pair set. This will happen, when at a particular input we want the $\mathcal{GP}$ regression value to be close to its positive target regression value $y(\mathbf{x})$ and far from its negative target regression value $\bar{y}(\mathbf{x})$.

### 3.2 FUNCTIONAL REPRESENTATION OF NEGATIVE DATAPAIRS

For our framework, we first get a functional representation of the negative datapairs. We define a Gaussian distribution around each of the negative datapair, $q(\bar{y}|\bar{\mathbf{x}}) \sim \mathcal{N}(\bar{y}(\bar{\mathbf{x}}), \sigma_{\text{neg}}^2)$, with mean equal to the negative target value $\bar{y}(\mathbf{x})$ and $\sigma_{\text{neg}}^2$ is the variance which is a hyperparameter. The Gaussian blob can also be thought of as the area of influence for the negative datapair with the variance $\sigma_{\text{neg}}$ indicating the spread of its influence.

### 3.3 $\mathcal{GP}$-NC REGRESSION FRAMEWORK

The aim of our $\mathcal{GP}$-NC regression framework is to simultaneously fit the $\mathcal{GP}$ regression on the positive datapairs ($X$) and avoid the negative datapairs ($\bar{X}$) (*i.e.*, using them as negative constraints (NC)). The former is achieved by maximizing the marginal likelihood given in the Eq. (4). To avoid

Algorithm describing the $\mathcal{GP}$-NC regression. We alternatively update between the negative log-likelihood and KL divergence term with respect to the kernel parameters $\theta$. For different $\mathcal{GP}$ methods we can appropriately plug-in the log-likelihood term (NLL).

---

**Algorithm 1:** Training of $\mathcal{GP}$-NC

**Input** : Datapairs $\{X, \mathbf{y}\}^{+}, \{\bar{X}, \bar{\mathbf{y}}\}^{-}$
**Parameters** : $\mathcal{GP}$ Kernel Parameters '$\theta$'
**Hyperparameters :** $\sigma_{\text{neg}}, \lambda$
**while** *until convergence* **do**
$\quad$ NLL = - $p(\mathbf{y}|\theta, X)$;
$\quad \theta \leftarrow$ minimize (NLL);
$\quad \text{KL}_{\text{div}} = \lambda \cdot \log D_{\text{KL}} \left( p\left(\hat{\mathbf{y}}|\theta, \bar{X}\right) || \mathcal{N}\left(\bar{\mathbf{y}}, \sigma_{\text{neg}}^2\right)\right)$;
$\quad \theta \leftarrow$ maximize $(\text{KL}_{\text{div}})$
**end**

---

the negative datapairs, we want our $\mathcal{GP}$ model to adjust its distribution curve so that while drawing samples from the predictive $\mathcal{GP}$ distribution, its values do not lie in the influence region of the negative datapairs. To this end, we propose to fit the $\mathcal{GP}$ regression model on the positive datapairs along with maximizing the Kullback-Leibler (KL) divergence between the distributions of the $\mathcal{GP}$ regression model and the Gaussian distributions defined over the negative datapairs.

Thus, mathematically, we want to maximize the following KL divergence given by

$$\Delta = \arg \max_{\theta} D_{\text{KL}}(p(\mathbf{y}|\theta, \bar{X})\|q(\bar{\mathbf{y}}|\bar{X})) \tag{5}$$

We chose to maximize the $D_{\text{KL}}$ term in the $p \to q$ direction, as this fixes the negative datapairs distribution $q(\bar{\mathbf{y}}|X)$ as the reference probability distribution. Now, since the KL divergence is an unbounded distance metric, the following section describes a practical workaround to maximize it.

### 3.3.1 Maximizing KL divergence using the logarithm trick

Eq. (5) is increasing the distance between the $\mathcal{GP}$ distribution and the negative datapairs distribution by maximizing the KL divergence. However, KL divergence is an unbounded function, *i.e.*, $D_{\text{KL}} \in [0, \infty)$. Implementing the $D_{\text{KL}}$ divergence directly in the form of Eq. (5) can create problems for the gradient updates and convergence.

We also want to maximize the marginal log-likelihood Eq. (4) and the $\Delta$ terms simultaneously. This raises a problem of mismatch in the magnitude of scale for the marginal log-likelihood term and $D_{\text{KL}}$ divergence term as the values of $D_{\text{KL}}$ divergence will be significantly higher. Thus, the gradient update would be dominated by $\Delta$ term. In essence, the model would fixate more on avoiding the negative datapairs than fitting the curve on the positive datapairs. We also observed this empirically. Hence, to suppress the gradient update from the $\Delta$ term, we encapsulate Eq. (5) in a logarithmic function. This turns out to be beneficial in multiple ways. Firstly, maximizing the $D_{\text{KL}}$ term is equivalent to maximizing the $\log(D_{\text{KL}})$, as $\log$ is an monotonically increasing function. Secondly, and more importantly, the scale of magnitude of $\Delta$ term becomes equivalent to the scale of magnitude for the marginal log-likelihood term which makes the convergence stable.

### 3.3.2 $\mathcal{GP}$-NC: Learning and model selection

We apply a $\log$ function to the $D_{\text{KL}}$ given in Eq. (5) and write the combined objective function for our $\mathcal{GP}$-NC regression,

$$\mathcal{L}(\theta) = \arg \min_{\theta} \left[ - \log p(\mathbf{y}|\theta, X) - \lambda \log D_{\text{KL}}(p(\mathbf{y}|\theta, \bar{X})\|q(\bar{\mathbf{y}}|\bar{X})) \right] \tag{6}$$

where $p(\mathbf{y}|\theta, X)$ is the marginal log-likelihood term that represents the model to be fitted on the observed datapoints. The parameter $\lambda$ is the tradeoff hyperparameter between curve fitting and avoidance of the negative datapoints, or how relaxed can the negative constraints be.

We already know the analytical form of the log-likelihood term from Eq. (4). We now focus on the $\log(D_{\text{KL}})$ term. Since, both the likelihood $p(\mathbf{y}|\theta, \bar{X})$ and negative datapair distributions $q(\bar{\mathbf{y}}|\bar{X})$ are modeled using Gaussians, we can simply use the analytical form of KL divergence between any two Gaussian distributions given by,

$$D_{\text{KL}}(p, q) = \log \frac{\sigma_2}{\sigma_1} + \frac{\sigma_1^2 + (\mu_1 - \mu_2)^2}{2\sigma_2^2} - \frac{1}{2} \tag{7}$$

here $p, q$ are Gaussian distributions defined as $\mathcal{N}(\mu_1, \sigma_1)$ and $\mathcal{N}(\mu_2, \sigma_2)$ respectively. The $D_{\mathrm{KL}}$ term is adjusting the mean and variance of the likelihood $p\left(Y|\theta, \bar{X}\right)$ with respect to the fixed blobs of Gaussian distributions around the negative datapairs. Specifically, we can consider the distribution '$p \equiv \mathcal{N}(\mu_1, \sigma_1) \equiv p(\mathbf{y}|\theta, \bar{X})$' and '$q \equiv \mathcal{N}(\mu_2, \sigma_2) \equiv q(\bar{\mathbf{y}}|\bar{X})$' in Eq. 7. Now, if we refer Eq. 2, $\mu_1 = E[\mathbf{f}_*]$ and $\sigma_1 = \mathrm{cov}(\mathbf{f}_*)$ which contain the parameters $\theta$ of the $\mathcal{GP}$ that are optimized. $\mu_2, \sigma_2$ correspond to the hyperparameters of the Gaussian distribution representing the negative datapairs and are constant. Algorithm (1) gives an overview of the training of $\mathcal{GP}$-NC regression.

*A **note** on the difference between the $\mathcal{GP}$-NC regression and general classification settings:* In the case of $\mathcal{GP}$-NC regression, the boundary for every negative datapair is optimized independent of each other. In classification settings, all the negative points belong to a class and they jointly affect the decision boundary of the $\mathcal{GP}$ for their class.

### 3.4    SPARSE GAUSSIAN PROCESSES WITH NEGATIVE DATAPOINTS

In Appendix A, we show that it is straightforward to modify the class of scalable and Sparse $\mathcal{GP}$ regression models to account for the negative datapairs in their formulation. In particular we review the SVGP model by (Hensman et al., 2013), which is a popular scalable implementation of $\mathcal{GP}$s. We also investigate a recent parametric Gaussian Process regressors (PPGPR) method by (Jankowiak et al., 2019). We evaluate the performance of these methods with our $\mathcal{GP}$-NC framework in the experiments section.

## 4    RELATED WORKS

*Classical $\mathcal{GP}$*: To the best of our knowledge, the classical $\mathcal{GP}$ regression introduced in (Rasmussen, 2006) and many subsequent works primarily focus on positive datapairs for curve fitting. Even with the absence of the concept of negative datapairs, $\mathcal{GP}$ regression methods have been widely used for obstacle-aware navigation task which is one of the relevant applications to evaluate our $\mathcal{GP}$-NC framework.

*$\mathcal{GP}$s for navigation*: $\mathcal{GP}$s are extensively used in the field of navigation and often are a component of path planning algorithms. (Ellis et al., 2009) used $\mathcal{GP}$ regression in modeling the pedestrian trajectories by using positive datapairs. (Aoude et al., 2013) used heuristic based approach over $\mathcal{GP}$ regression to incorporate dynamic changes and environmental constraint in the surroundings. Their solution, named RR-GP, builds a learned motion pattern model by combining the flexibility of $\mathcal{GP}$ with the efficiency of RRTReach, a sampling-based reachability computation. Obstacle trajectory GP predictions are conditioned on dynamically feasible paths identified from the reachability analysis, yielding more accurate predictions of future behavior. (Goli et al., 2018) introduced the use of $\mathcal{GP}$ regression for long-term location prediction for collision avoidance in Connected Vehicle (CV) environment. The $\mathcal{GP}$s are used to model the trajectory of the vehicles using the historical data. The collected data from vehicles together with GPR models received from infrastructure are then used to predict the future trajectories of vehicles in the scene. (Meera et al., 2019) designed an Obstacle-aware Adaptive Informative Path Planning (OA-IPP) algorithm for target search in cluttered environments using UAVs. This method uses $\mathcal{GP}$ to detect the obstacles/target, which the UAV gets by marking dense number of points (positive datapairs) around the obstacles. (Hewing et al., 2020; Yuan & Kitani, 2019) are some of the works using sampling based techniques for trajectory prediction. (Choi et al., 2015) is one of the work which tries to incorporate the concept of negative datapairs in classical $\mathcal{GP}$ construct by introducing a *leveraged parameter* in kernel function. The authors demonstrate that having the ability to incorporate negative targets increases the efficiency of trajectory predictions. However, this approach fundamentally differs from ours in terms of incorporation of negative datapairs ours try to maximize the KL-Divergence between $\hat{y}(x)$ and $\bar{y}(x)$ while theirs utilizes additional *leveraged parameter* in the kernel function. Besides, our approach is more scalable as the size of the covariance matrix doesn't increase to incorporate the negative datapairs.

*Scalable $\mathcal{GP}$*: Naïve implementation of $\mathcal{GP}$ regression is not scalable for large datasets as the model selection and inference of $\mathcal{GP}$ requires a cubic order of computations $\mathcal{O}(n^3)$ and $\mathcal{O}(n^2)$ of storage. Since the $\mathcal{GP}$-NC framework is quite generic and can work for various scale $\mathcal{GP}$ methods, we want to highlight few of these methods. (Hensman et al., 2013; Dai et al., 2014; Gal et al., 2014) are some of the well known scalable methods suitable for our framework as they use stochastic gradient descent

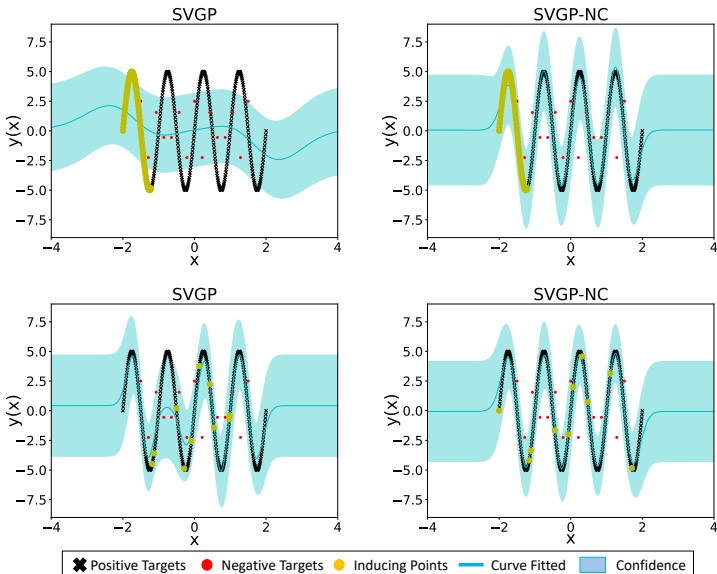

Figure 2: Visualizing $\mathcal{GP}$-NC regression framework: The figures compare how the SVGP regression fits using the classical $\mathcal{GP}$ framework (left) vs the $\mathcal{GP}$-NC framework (right). The aim is to fit the regression targets marked in '**black**' (positive datapairs) and avoid the targets marked in 'red' (negative datapairs). The classical $\mathcal{GP}$ framework only uses the positive datapairs whereas our proposed $\mathcal{GP}$-NC framework uses both the positive and negative datapairs for fitting the regression curve. The points in 'yellow' are the inducing points used to fit the $\mathcal{GP}$. We used two inducing points setting. Top figures: locations of inducing points were taken at the start of curve. Bottom figures: we randomly sampled the inducing points from the whole range of training inputs. For $\mathcal{GP}$-NC framework (right), hyper-parameters were selected as $\lambda = 0.1$ and $\sigma_{\text{neg}} = 1.2$

for optimization. Furthermore, recent works (Wilson & Nickisch, 2015; Wilson et al., 2015; 2016) have improved scalability by reducing the learning to $\mathcal{O}(n)$ and test prediction to $\mathcal{O}(1)$ under some assumptions.

*Negative datapairs in other domains*: The concept of negative datapairs has been extensively utilized in the self-supervised learning. Applications include learning word embeddings (Mikolov et al., 2013; Mnih & Kavukcuoglu, 2013), image representations (He et al., 2020; Misra & Maaten, 2020; Feng et al., 2019), video representations (Sermanet et al., 2018; Fernando et al., 2017; Misra et al., 2016; Harley et al., 2020), etc. In these works negative and positive samples are created as pseudo labels to train a neural network to learn the deep representations of the inputs.

## 5 EXPERIMENTS

We compared various $\mathcal{GP}$ regression models in their classical form (using only positive datapairs) with their corresponding $\mathcal{GP}$-NC regression models where we used our negative constraints framework. We used Negative Log-likelihood (NLL) and Root Mean Squared Error (RMSE) as our evaluation metrics. We compared our framework on a synthetic dataset and six real world datasets. Throughout our experiments, we found that for every $\mathcal{GP}$ model, the $\mathcal{GP}$-NC regression framework outperforms its corresponding classical $\mathcal{GP}$ regression setting. We used GPytorch (Gardner et al., 2018) to implement all the $\mathcal{GP}$ (ours + baselines) models. We use zero mean value and the RBF kernel for $\mathcal{GP}$ prior for all of the models unless mentioned otherwise.

### 5.1 SYNTHETIC DATASET: VISUALIZING THE $\mathcal{GP}$-NC REGRESSION FRAMEWORK

We aim to visualize the $\mathcal{GP}$-NC regression framework using a toy dataset. We sampled 400 positve datapairs from a sinusoidal function and randomly sampled 15 negative datapairs as represented by Fig. 2. We trained a sparse SVGP model to regress a curve on the positive datapair using the classical $\mathcal{GP}$ framework and the one with negative constraints $\mathcal{GP}$-NC . For the top figures of Fig. 2, we trained SVGP with 80 inducing points, all at the starting location of training input range. For the bottom figures of Fig. 2, we randomly sampled 10 inducing points from the range of training inputs. SVGP with a constant mean and a RBF kernel for the $\mathcal{GP}$ prior was used. After training the SVGP model in both settings for 100 epochs we obtain the curves as depicted by the figures on the

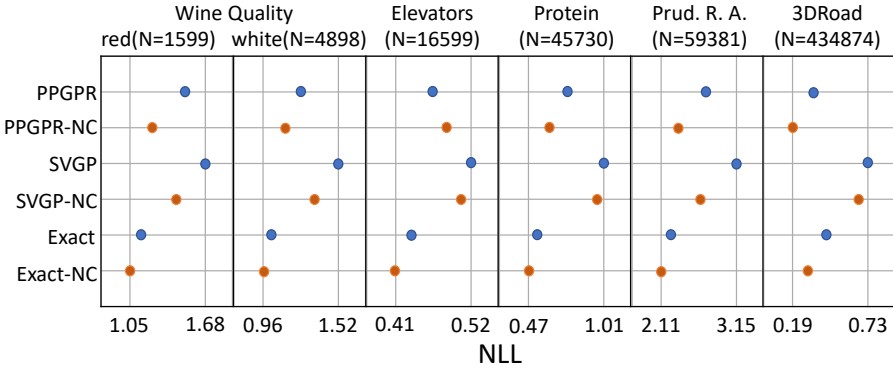

Figure 3: Comparison on real world data: We plot test negative log-likelihoods (NLL) for 6 univariate regression datasets (lower is better). Results are averaged over 10 random train/test/valid splits.

Table 1: **Runtime comparison** of the classical $\mathcal{GP}$ and $\mathcal{GP}$-NC frameworks which includes negative datapairs on different datasets. $\Delta_t$ is the runtime difference of the $\mathcal{GP}$ model in $\mathcal{GP}$-NC framework vs the classical $\mathcal{GP}$ framework. We used GPU accelerated $\mathcal{GP}$ implementation of GPyTorch library.

| Datasets | Size of data | Type of Target Variable | $\Delta_t$ Exact $\mathcal{GP}$ | $\Delta_t$ sparse SVGP |
|---|---|---|---|---|
| Wine quality - red | 1599 | Discrete | 21ms | 3.1s |
| Wine quality - white | 4898 | Discrete | 40ms | 3.7s |
| Elevators | 16599 | Continuous | 5s | 34s |
| Protein | 45730 | Continuous | 50s | 37s |
| Prudential | 59381 | Discrete | 1.2 s | 34s |
| 3DRoad | 434874 | Continuous | 2208s | 292s |

left-side in Fig. 2. The inability to incorporate the information provided by the negative datapairs in the classical $\mathcal{GP}$ construct hinders its ability to fit the data well as patently visible in the left figure. Mean and predictive variance are not only losing out on some positive datapairs but are also engulfing the negative datapairs in the confidence region which is undesirable.

Our $\mathcal{GP}$-NC framework re-calibrates the curve by integrating the information provided by the negative datapairs as seen in the right-hand side figures of Fig 2. As evident from the figure, the additional information from a few negative datapairs helps the model to fit better to the positive datapairs in addition to avoiding most of the negative datapairs. We can tune the values of $\lambda$ given in Eq. (6) to balance between the weightage given by the $\mathcal{GP}$ to positive and negative datapairs. Decreasing the value of $\lambda$ results in reduction of influence of the negative datapairs. Notice that curved learned by our approach even with sub-optimal inducing points.

## 5.2 Trajectory prediction using $\mathcal{GP}$-NC regression framework

We want to model an agent's trajectory using a $\mathcal{GP}$ regression model such that it takes the agent's present location $(x, y)$ as input and predicts agents next location $(\hat{x}, \hat{y})$. For this set of experiments, we synthesize a 2d-virtual traffic scene given by Fig. 4. Furthermore, the road contain pitfalls, roadblocks, accidents, etc. that need to be avoided are represented as red diamonds in the figure. We designate these targets as negative targets that are to be avoided for ensuring the safety of traffic. There are a total number of 10 negative datapairs present in the scene. Next, We sample 250 observed co-ordinates on the 2d virtual path for modeling the future trajectory of the agent.

We trained a classical SVGP model and ours SVGP-NC model to predict the trajectory of agent. For both the models we utilize a constant mean and an RBF kernel for the $\mathcal{GP}$ prior. Both the models were trained for 100 epochs. It can be observed from the Fig. 4.a that the classical $\mathcal{GP}$ framework lacks the ability to incorporate negative datapairs, which results in a loosely fitted $\mathcal{GP}$ model. On the other hand, when trained with an additional constraints given by negative datapairs $\mathcal{GP}$-NC fits a tighter curve on the observed datapairs and avoiding all the negative datapairs as shown in Fig 4.b.

Moreover, in these set of experiments it is easy to demonstrate the impact of the $\lambda$ values from Eq. (6) on the $\mathcal{GP}$ regression. Decreasing the value of $\lambda$ results in reduction of influence of the negative

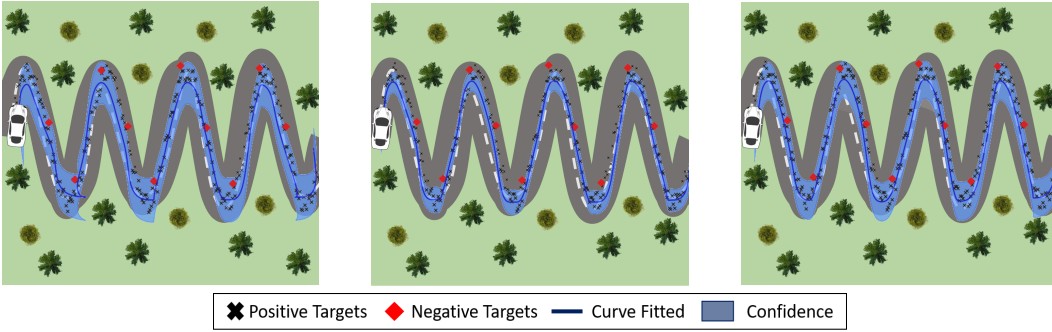

Figure 4: Trajectory prediction with $\mathcal{GP}$-NC regression framework: The figures compare trajectory prediction in a 2D-virtual environment using the classical $\mathcal{GP}$ framework (a) vs. the $\mathcal{GP}$-NC framework (b,c). The car is navigating through the forest and our aim is to avoid the roadblocks marked in 'red' while maintaining the car's proximity to the '**black**' trajectory markers. The classical $\mathcal{GP}$ framework only uses the positive datapairs whereas our proposed $\mathcal{GP}$-NC framework uses both the positive & negative datapairs for prediction of agent's trajectory. (b) depicts the $\mathcal{GP}$-NC framework with the hyper parameter $\lambda = 1$ (c) depicts the $\mathcal{GP}$-NC framework with the hyper parameter $\lambda = 0.1$

datapairs as can be observed from Fig 4.b vs. Fig 4.c. However, it can be observed by juxtaposing both the figures (Fig 4.a and Fig 4.c) that predicted mean trajectory by classical $\mathcal{GP}$ and $\mathcal{GP}$-NC with $\lambda = 0.1$, the latter fits better to the black trajectory markers. Hence, for saftey critical applications like navigation $\mathcal{GP}$-NC is superior than the classical $\mathcal{GP}$ in incorporating negative constraints.

## 5.3 REALWORLD DATASETS

We evaluated our $\mathcal{GP}$-NC framework on six real world datasets with the number of datapoints ranging from $N \sim 1500, 5000, 15000, 50000, 450000$ and the number of input dimension $d \in [3, 127]$. Among the six datasets five of them are from UCI repository (Dua & Graff, 2017) (Wine quality - red, white, Elevators, Protein, and 3DRoad), while the sixth one is from Kaggle Competition (Prudential life insurance risk assessment). These datasets consists of two different kind of prediction/regression variables namely discrete variable and continuous variable. For discreet variable the value of elements lies between certain range, *i.e.*, integer values lie between $[0, 10]$. For continuous variable, the value of target regression can be any real number. Datasets (Prudential risk assessment, Wine quality - red, white) are all discrete target variable datasets while datasets (Elevators, Protein and 3DRoad) have continuous target variables.

***Random shuffling technique*** *for creating negative datapairs for* $\mathcal{GP}$-*NC*: As we are only given positive regression target values in these datasets, we create pseudo-negative regression targets by randomly shuffling the labels and pairing them with the inputs to create negative datapairs. This generates a set of valid datapairs as given the input $\mathbf{x}$ only $y(\mathbf{x})$ can be associated as true regression value/label, we can assume whatever label we get by random shuffling as a negative target.

We compared our model against the standard baselines of Exact $\mathcal{GP}$ (Gardner et al., 2019; Wang et al., 2019), and Sparse $\mathcal{GP}$ methods like SVGP (Hensman et al., 2013) and PPGPR (Jankowiak et al., 2019). For training the sparse methods, we used 1000 inducing points. We trained the models using Adam optimizer with a learning rate of 0.1 for 400 epochs on each dataset. For $\mathcal{GP}$-NC framework, we used 200 negative datapairs. We maintained consistency in all the models in all terms of maintaining a constant mean and RBF kernel for a $\mathcal{GP}$ prior.

Fig. 3 compares negative log likelihood values of various $\mathcal{GP}$ regression methods, in both classical and $\mathcal{GP}$-NC frameworks, on all six real datasets. The orange dots represents our methods while the blue dots in the plots depict the baselines. It can be observed that $\mathcal{GP}$-NC framework outperforms the classical $\mathcal{GP}$ framework. Methods like SVGP - NC and Exact $\mathcal{GP}$ - NC performs on an average 0.2 nats better than baseline SVGP and Exact $\mathcal{GP}$ respectively. It is interesting to note that including negative datapairs by the 'random shuffling technique' is quite effective and we can observe gains in terms of model performance.

Table 1 compares runtime difference between the classical $\mathcal{GP}$ and the $\mathcal{GP}$-NC to see how the additional KL divergence term that accounts for negative datapairs affects the runtime of $\mathcal{GP}$-NC framework. We train all the $\mathcal{GP}$ models for 50 epochs and measure the average excess time ($\Delta_t$)

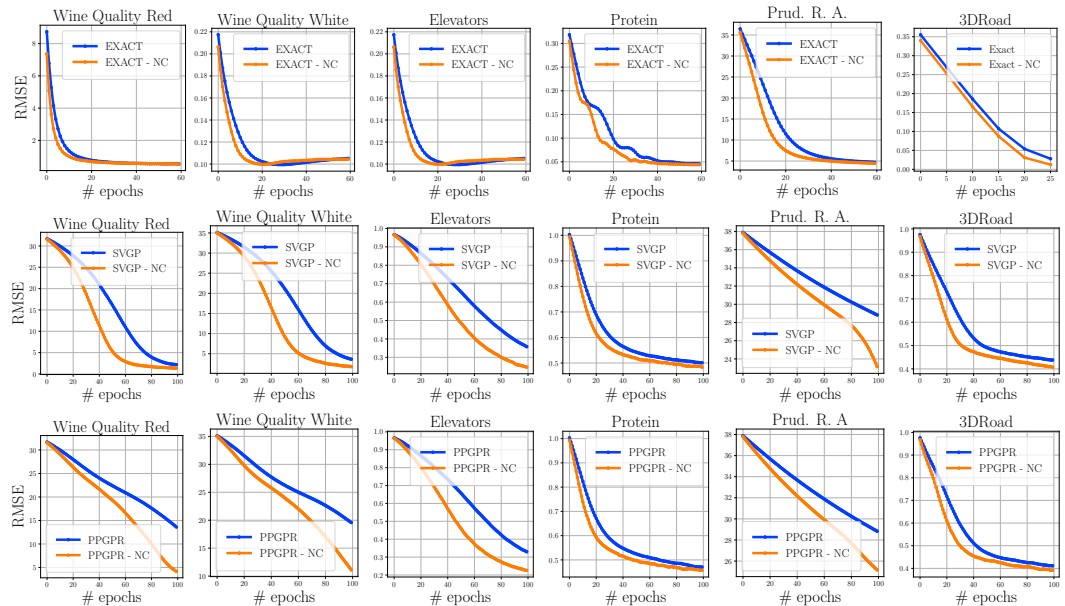

Figure 5: RMSE plots on real world data (top - **Exact** $\mathcal{GP}$; middle - **SVGP**; bottom - **PPGPR**): Plots show the test RMSE for six univariate regression datasets (lower is better). Models are fitted by using cross validation on training data. Convergence of $\mathcal{GP}$-NC framework is consistently faster than its classical $\mathcal{GP}$ counterpart for all the models.

value over 10 runs. It can be observed from Table 1 that the $\Delta_t$ depends on the size of the dataset and also on the type of target regression variable. For the Exact $\mathcal{GP}$ and discrete target variables setting the $\Delta_t$ does not increase much with size of the dataset, however for continuous target variable there is a considerable amount of increase. For SVGP model the increase in $\Delta_t$ can be attributed to the size of the dataset. Overall, the average increase in training is not very significant. Thus, our experiments indicate that the added penalty term to likelihood term in Eq. (6) does not significantly affect the scalability of current scalable $\mathcal{GP}$ architectures.

Figure 5 shows the RMSE plots for Exact $\mathcal{GP}$, SVGP, and PPGPR models with the classical $\mathcal{GP}$ juxtaposed on to the models with the negative constraint $\mathcal{GP}$-NC framework. It can be observed from the plots that our $\mathcal{GP}$-NC models converges faster, and better, than the classical $\mathcal{GP}$ models for the six univariate real world datasets. In addition, it is interesting to note that as the size of the data increases, the convergence curve of the $\mathcal{GP}$-NC model becomes steeper.

## 6 CONCLUSION

We presented a novel and generic Gaussian Process regression framework $\mathcal{GP}$-NC that incorporates negative constraints along with fitting the curve for the positive datapairs. Our key idea was to assume small blobs of Gaussian distribution on the negative datapairs. Then, while fitting the $\mathcal{GP}$ regression on the positive datapairs, our $\mathcal{GP}$-NC framework simultaneously maximizes the KL divergence from the negative datapairs. Our work highlights the benefits of modeling the negative datapairs for $\mathcal{GP}$s and our experiments support the effectiveness of our approach. We hope that this successful realization of the concept of negative datapairs for $\mathcal{GP}$ regression will be useful in variety of applications.

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

## A $\mathcal{GP}$-NC FOR SCALABLE $\mathcal{GP}$ METHODS

We can replace the NLL term in Algorithm (1) by the log likelihood of the different scalable $\mathcal{GP}$ methods. We have a scalable implementation of the $D_{\text{KL}}$ update, so the entire Algorithm scales well with the input data size. It is straightforward to plug-in the class of scalable and Sparse $\mathcal{GP}$ regression models in the likelihood term of Algorithm (1) to account for the negative datapairs in their formulation. In particular we review the SVGP model by (Hensman et al., 2013), which is a popular scalable implementation of $\mathcal{GP}$s. We also investigate a recent parametric Gaussian Process regressors (PPGPR) method by (Jankowiak et al., 2019). In this section, we follow the notations given in their respective research works and give their derivations of the log likelihood function here for the sake of completeness.

### A.1 SVGP REGRESSION MODEL

(Hensman et al., 2013) proposed the Scalable Variational GP (SVGP) method. The key technical innovation was the development of inducing point methods which we now review. By introducing inducing variables $\mathbf{u}$ that depend on variational parameters $\{\mathbf{z}_m\}_{m=1}^M$, where $M = \dim(\mathbf{u}) \ll N$ and with each $\mathbf{z}_m \in \mathbb{R}^d$, we augment the GP prior as follows:

$$p(\mathbf{f}|X) \rightarrow p(\mathbf{f}|\mathbf{u}, X, Z)p(\mathbf{u}|Z)$$

We then appeal to Jensen's inequality and lower bound the log joint density over the targets and inducing variables:

$$
\begin{aligned}
\log p(\mathbf{y}, \mathbf{u}|X, Z) &= \log \int d\mathbf{f} p(\mathbf{y}|\mathbf{f})p(\mathbf{f}|\mathbf{u})p(\mathbf{u}) \\
&\geq \mathbb{E}_{p(\mathbf{f}|\mathbf{u})}\left[\log p(\mathbf{y}|\mathbf{f}) + \log p(\mathbf{u})\right] \\
&= \sum_{i=1}^N \log \mathcal{N}(y_i | \mathbf{k}_i^T K_{MM}^{-1}\mathbf{u}, \sigma_{\text{obs}}^2) - \frac{1}{2\sigma_{\text{obs}}^2}\text{Tr}\, Kt_{NN} + \log p(\mathbf{u})
\end{aligned}
\tag{8}
$$

where $\mathbf{k}_i = k(\mathbf{x}_i, Z)$, $K_{MM} = k(Z, Z)$ and $Kt_{NN}$ is given by

$$Kt_{NN} = K_{NN} - K_{NM}K_{MM}^{-1}K_{MN} \tag{9}$$

with $K_{NM} = K_{MN}^{\text{T}} = k(X, Z)$. The essential characteristics of Eqn. 8 are that: i) it replaces expensive computations involving $K_{NN}$ with cheaper computations like $K_{MM}^{-1}$ that scale as $\mathcal{O}(M^3)$; and ii) it is amenable to data subsampling, since the log likelihood and trace terms factorize as sums over datapoints $(y_i, \mathbf{x}_i)$.

#### A.1.1 SVGP LIKELIHOOD FUNCTION

SVGP proceeds by introducing a multivariate Normal variational distribution $q(\mathbf{u}) = \mathcal{N}(\boldsymbol{m}, S)$. The parameters $\boldsymbol{m}$ and $S$ are optimized using the ELBO (evidence lower bound), which is the expectation of Eqn. 8 w.r.t. $q(\mathbf{u})$ plus an entropy term term $H[q(\mathbf{u})]$:

$$
\begin{aligned}
\mathcal{L}_{\text{svgp}} &= \mathbb{E}_{q(\mathbf{u})}\left[\log p(\mathbf{y}, \mathbf{u}|X, Z)\right] + H[q(\mathbf{u})] \\
&= \sum_{i=1}^N \left\{\log \mathcal{N}(y_i|\mu_{\mathbf{f}}(\mathbf{x}_i), \sigma_{\text{obs}}^2) - \frac{\sigma_{\mathbf{f}}(\mathbf{x}_i)^2}{2\sigma_{\text{obs}}^2}\right\} - D_{\text{KL}}(q(\mathbf{u})|p(\mathbf{u}))
\end{aligned}
\tag{10}
$$

where KL denotes the Kullback-Leibler divergence, $\mu_{\mathbf{f}}(\mathbf{x}_i)$ is the predictive mean function given by $\mu_{\mathbf{f}}(\mathbf{x}_i) = \mathbf{k}_i^T K_{MM}^{-1}\boldsymbol{m}$ and $\sigma_{\mathbf{f}}(\mathbf{x}_i)^2 \equiv \text{Var}[f_i|\mathbf{x}_i] = Kt_{ii} + \mathbf{k}_i^T K_{MM}^{-1}SK_{MM}^{-1}\mathbf{k}_i$ denotes the latent function variance.

$\mathcal{L}_{\text{svgp}}$, which depends on $\boldsymbol{m}, S, Z, \sigma_{\text{obs}}$ and the various kernel hyperparameters $\theta$, can then be maximized with gradient methods. We refer to the resulting GP regression method as SVGP.

### A.2 PPGPR-NC REGRESSION MODEL: LIKELIHOOD FUNCTION

Jankowiak et al. (2019) recently proposed a parametric Gaussian Process regressors (PPGPR) method. We defer the reader to Section (3.2) of their paper for details about their likelihood function.

