# OpenReview forum: "Learning What Not to Model: Gaussian Process Regression with Negative Constraints"
_ICLR.cc/2021/Conference — Reject_

### Official Review · AnonReviewer2 · 2020-10-24
**Not sufficiently motivated**

**Rating:** 3
**Confidence:** 4

**Review:**

SUMMARY

This paper introduces GP-NC, a new methodology that allows Gaussian processes to stay far from certain data points (referred to as negative datapairs). These negative datapairs complement the standard training points to be fitted. The authors show enhanced performance when using this technique with standard GPs, SVGP and PPGPR.

###################################################

PROS

1) I am not aware of previous works in the GP literature dealing with this scenario, i.e. data pairs to be avoided during training.

####################################################

CONS

1) I think this work is not sufficiently motivated. The authors claim that the brand-new negative datapairs can be useful for modelling navigation problems with GPs. However, this type of problem is not addressed in the experimental section. I would expect to see how the proposed method compares to the previous approaches that have modelled the navigation problem with GPs.

2) In fact, due to the lack of real problems that require the presented technique, in the experimental section they just simulate some negative datapairs (following the so-called random shuffling technique). Moreover, this simulation technique is pretty poor in my view, since different inputs could have very similar outputs, which may lead to a simulated datapair almost identical to a standard training datapair.

3) Some arguments are too vague/ambiguous. For instance, the equivalence between the scale of magnitude of both terms in eq. (5). It is stated that, by using the so-called logarith trick, both terms have the same "scale of magnitude". How is that defined? and how can the equivalence be proved? Or the sentence that says "A Gaussian process is a Bayesian non-parametric approach that fits Gaussian distributions to functions".

4) Low quality writing in general. I have come across many typos and sentences that are not well written. See e.g.:
* caption of Fig. 1 "is been given".
* after eq. (5) "minimize",
* "Hewing et al. (2020) are some of the works using sampling based techniques for trajectory prediction"
* "We trained a sparse SVGP model to regress a curve on the using the classical GP framework and the one with negative constraints GP-NC"
* Caption of Figure 4: it says five datasets, but there are six
* Many references are missing the journal/conference/arxiv reference. Format not homogeneous across references.


############################################

AFTER THE REBUTTAL

I have decided to keep my score unchanged. The authors include a new "road navigation experiment" in Section 5.2. However, I have several concerns about it:
* This is a synthetic experiment that the authors have created themselves. Therefore, it does not address my concern on seeing actual real-world problems that can be solved with this approach.
* The results of GP-NC (the proposed approach) and GP are very similar. I don't really see that the predictive mean of GP-NC avoids the "negative samples" more than the standard GP.
* In fact, the design of this experiment is not clear to me. They say at the beginning that they use the present location ($x,y$) to predict the next location ($\hat x, \hat y)$. How do they model these two outputs? How do they obtain the plotted error bars for this model? This is not sufficiently explained. My impression is that they are just fitting a standard 1-dimensional GP being the input the x-dimension and the output the y-dimension.

---

> ### Author Response · Authors · 2020-11-23
> **Additional navigation experiments added to the revised paper**
>
> We thank the reviewer for carefully reading our work and acknowledging the novelty of the work.  We address the cons listed by the reviewer below:
> 1. We do believe that modeling negative datapairs is an important problem faced by researchers and engineers working on safety-critical problems.  We have added a road navigation experiment in our updated draft to partly address your concern.
> 2. Regarding the random shuffling technique, we used extreme precautions that no negative targets of the simulated datapair are close to the positive targets of existing standard datapair. Hence, during the experiments, it can be observed that GPNC performs better than the standard GP.
> 3. We have worked on your comments to raise the quality of writing in the updated draft.
>
>
> **Updates to the paper:**
> * Added a new set of experiments to address concerns over the GP navigation problems. (Sec. 5.2).
> * Improved presentation in the updated paper.

---

### Official Review · AnonReviewer4 · 2020-10-27
**Incorporating negative constraints into Gaussian process regression**

**Rating:** 6
**Confidence:** 2

**Review:**

Summary:
This paper incorporates information of obstacles to avoid (e.g robot navigation trajectory in the room where the robot has to avoid items such as furniture) into Gaussian process regression fit. They call the obstacles, negative datapairs and the rest of data, positive datapairs. The aim is to have a GP where the probability of passing through the negative datapairs is low. The proposed method is called the Gaussian process with negative constraints (GP-NC).

To be able to fit the GP regression to positive datapairs and avoid negative datapairs, they maximize the KL-divergence between the distributions of GP learned from positive datpairs $(p(y|\theta. X))$ and negative datapiars $(q(\hat{y}|\hat{X}))$ which will have a bound between $[0, \inf]$.
For being able to maximize the KL-divergence with the marginal log-likelihood they change the scale of KL-divergence to the log scale and a parameter $\lambda$. $\lambda$ is a tradeoff between curve fitting and avoidance of the negative datapoints. KL term has an analytical solution since both distributions are Gaussians.
They compare their method against SVGP (Hensman et al., 2013) and PPGPR (Jankowiak et al., 2019) and the exact GP.

Comments and questions:
The paper is well-written and easy to follow. This paper is also technically sound and to the best of my knowledge is novel and relevant to the community.

In figure two you have mentioned you sampled the inducing inputs randomly from the whole range of training inputs. Did you choose the same inducing inputs for the SVGP and SVGP-NC?

In the error calculation, standardized mean squared error (SMSE) could be a better choice than RMSR since the former incorporates variance information as well.

In the experiment section, all negative datapairs are synthetically made. I was wondering if you could apply your method to a dataset with real negative datapiars (like a robot in the room avoiding obstacles)?

In figure 3 (3DRoad) the approximate results are better than the exact GP. Is this because of the scale of the data and not being able to converge?

Miscellaneous comments:
In Table 1 row 2 should be wine quality - white

################### After the rebuttal ################

I thank the authors for addressing the issues that were raised. The paper is indeed addressing a very practical issue in the ML community.
After reading other reviewers' comments and concerns I decreased my score.

---

> ### Author Response · Authors · 2020-11-23
> **Paper updated; Experiments added to the updated paper**
>
> We want to thank the reviewer for thoughtful comments; especially for acknowledging the novelty of the approach.  We have incorporated your suggestions in our updated draft.
>
> **Inducing points**
>
> In Figure 2.c) and 2.d) we randomly selected 10 data points from the training data as inducing points and trained both the models. In doing so, we jointly learn the locations of the inducing points while fitting the curve. More details can be found in Appendix A1.
>
> **SMSE**
>
> We used standard practices from the GP literature for our evaluations. That said, we used RMSE error to emphasize the rate of convergence for comparison between classical GPs vs. GP-NC versions.
>
> **Navigation Experiments**
>
> We have updated our experiment section by adding a new set of navigation experiments. Please refer to Section 5.2 of our updated manuscript where we tried to address the concerns of R4 by applying our approach on simulated data where both the positive and negative datapairs occur naturally.
>
> **Misc**
>
> As the scale of the data increases, we use more approximations like structured kernel interpolation for exact GP which results in a decrease in performance to a certain extent. Lastly, we fixed the typos in our revised manuscript, as suggested. We again want to thank the reviewer for providing us with valuable feedback.
>
> **Updates to the paper:**
> * Added new set of experiments (Section 5.2)
> * Fixed typos

---

### Official Review · AnonReviewer3 · 2020-10-27
**Learning what not to model**

**Rating:** 3
**Confidence:** 4

**Review:**

Summary
--

This paper in concerned with Gaussian process regression under constraints that aim to discourage the model from learning certain values (negative constraints). These are called negative data pairs, and the authors propose an extension to the standard GP methodology to incorporate these constraints in the model. This is done by iteratively training a standard GP and maximising the KL between the GP and blobs of the negative data pairs.

The problem tackled in the paper is of relevance in fields such as spatial statistics and robotics, with possible applications also in more general ML tasks. The paper could be of interest to a narrow audience at ICLR. The presentation is rather clear and the main idea is easy to follow.


Concerns
--

1. My main concern is originality and novelty of the approach presented in the paper. The authors tackle an important practical problem (or limitation), but the presented contribution alone is small. The proposed approach for including negative data pairs is straightforward, and something that one might find in an application paper (where the main interest is in solving a task related to an application).

2. Limitations. The limitations of the approach should have been discussed in detail. Questions related to uni- vs. multimodality (splitting), bias influenced by the parameter \lambda, behaviour in multi-output (2d, like Fig. 1), would perhaps best have been illustrated with suitable simulated test cases and included as figures (including random samples from the GP posterior). Some of the issues are visible in Fig. 2, but these are not discussed in detail.

3. Experiments. The experiments do not appear convincing. Rather than artificially changing standard test data sets to fit your task setting, it would have been more interesting to see actual problems, where the model class would have been beneficial.

4. Presentation. Even if the method itself is easy to understand from how it is presented, the paper would deserve improvements in the presentation. As a practical suggestion: Background material could be presented more concisely, the methods section refined to be more concise, and the additional space used for improving experiments and discussion.

5. From how Alg. 1 now reads, I would interpret \lambda not to have any effect on the training. This should not be the case. Alg. 1 does not agree with the objective in Eq. (6).

6. Minor: Typo in Eq. (4).

---

> ### Author Response · Authors · 2020-11-23
> **Results of additional experiments added to the updated paper**
>
> We thank the reviewer for their comments. Below we address the concerns.
>
> 1. We believe that our work addresses a critical and important problem faced by the researchers while applying GP to problems in safety-critical applications, like navigation systems. We feel that the technique we proposed is scalable and our initial exploratory experiments support our claims. Nevertheless, we are constantly looking to improve our method, it will be helpful if you can suggest any possible additions to our work which we should include in our future versions.
> 2. We have added extra experiments in which we vary $\lambda$ to show their influence on trajectory prediction. Please refer to Section 5.2 of the updated manuscript.
> 3. $\lambda$ is a hyperparameter and it decides the balance between the 2 terms in Eq.(6). Please also refer to the newly added experiment on varying $\lambda$
> 4. We’ve fixed other requested typos.
>
> **Updates to the paper:**
> * Added new set of experiments (Section 5.2)
> * Updated [1] the related work section
> * Fixed typos

---

### Official Review · AnonReviewer1 · 2020-10-28
**Novel approach, but not a new problem in GP regression.**

**Rating:** 5
**Confidence:** 4

**Review:**

Summary
--------
The paper addresses Gaussian process regression (GPR) over 'positive' and 'negative' data points, such that the regression model fit the former and avoid the latter. The 'negative' data points in this work are each represented by individually distributed Gaussian random variables, with known parameters (mean vector and variance).
The authors propose a novel GPR loss (6) for optimizing GP model parameters $\theta$, striking a compromise between the regular GP loss and the KL divergence between the predictive GP distribution and the set of 'negative' data points. The approach is novel and an interesting take on the problem, and it has several beneficial properties which are investigated empirically.

Strong points
-------------
1. Compared to previous work on GPR with 'positive' and 'negative' data points, the proposed solution is elegant in its simplicity, and scalable for predictions (since the 'negative' data is only used during kernel parameter optimization). I especially like the scalability aspect, since it would allow the use of a otherwise prohibitively large amount of 'negative' examples. This is useful in navigation, for example in contextes where the boundaries of static obstacles are commonly represented by potential fields.
2. The experiments are fairly extensive and demonstrate several beneficial properties of the proposed method, such as fast learning convergence and compatibility with sparse GP methods.

Weak points
-------------
1. The problem of GPR with both 'positive' and 'negative' data points (and in the context of robot navigation) has been investigated before [1], where a non-stationary kernel function is proposed. The approach in [1] should be addressed, and compared with, at least qualitatively.
2. The experiments are not representing any situation where the distinction of 'positive' and 'negative' data points naturally occur. This makes it hard to assess how well the proposed method works for real problems of the kind the paper intends to address.
3. The technical clarity is varied:
     1. Equation (5) is incorrect? The two multivariate Gaussian distributions have different dimension (n vs m), and they do not share the same support ($X$ and $\bar X$ can differ). Maybe it should be "$p(y|\theta, \bar X)$". But is this then entirely compatible with the prior sentence: $\theta$ is in essence the "GP regression model learned from the positive data pairs" and yet it is stated that you maximize KL divergence between the GP regression model and the positive data pairs with respect to $\theta$? If $\theta$ is changed, then the first distribution in the KL divergence is no longer the GP model learned from the positive data points. I suggest that this section is made clearer and more precise. Equation (6) on the other hand corresponds to the stated proposal.
   2. It is not obvious to me how you make use of the re-parameterization trick proposed by (Kingma & Welling, 2013).
   3. Why do you not minimize the loss (6) directly, instead of first minimizing NLL and then maximizing KL at each iteration (Algorithm 1)?
   4. What value does $\lambda$ and $\sigma_{neg}$ have in the experiments?
4. The 'Random shuffling technique' is not at all clear to me.
   1. How can you get a set of *valid* data pairs (i.e. $x_i$ and $y_i$) by shuffling the labels ("$y_i$")?
   2. If randomly create pseudo-negative data points, especially using existing data points (that are either designated 'negative' or shuffled in some way to produce new input-output pairs as 'negative'), then isn't it likely given real data sets that these data points will be in conflict (e.g. overlap) with other 'positive' data points? If so, what do you mean with that these are *valid* and what does this mean for the experiment (more details on the limitations of the experiment and interpretations of the results should be present in the paper)?

Reason for score
----------------
Although an interesting take on the problem, the paper in its current form is lacking technical clarity, comparisons to directly related works, and the evaluation is weak in that it does not consider actual instances of the addressed problem in the empirical evaluation.

Minor comments
--------------
"to model the uncertainty by providing the predictive variance" -> "to model [target] uncertainty by providing predictive variance"

"in both the positive and negative data pair." -> "in both the positive and negative data pair set."

Page 4: "$P(y|\theta, X)$" -> "$p(y|\theta, X)$"

"loglikelihood" -> "log likelihood"

--------------

[1] S. Choi, E. Kim, K. Lee and S. Oh, "Leveraged non-stationary Gaussian process regression for autonomous robot navigation," 2015 IEEE International Conference on Robotics and Automation (ICRA), Seattle, WA, 2015, pp. 473-478, doi: 10.1109/ICRA.2015.7139222.

Edit: Markdown problem with nested lists

Edit: Upgraded rating from 4 to 5

-----
Thank you for addressing most of my concerns.

I have increased my rating, but the paper still needs some work in my opinion. More specifically,
* a more thorough comparison with [1], e.g. by including the quantitative comparison you were working on.
* an improved representative experiment from the problem domain. The newly added one is a bit simplistic and not motivated enough. Although stated to be 2D, it is (as far as I can tell) indistinguishable from a 1D regression task? How do you handle the 2D output in GPR? How are the data points generated that are used as training data, and what could motivate such a generation of data in a plausibly real setting?

---

> ### Author Response · Authors · 2020-11-23
> **Paper revised; added new experiments**
>
> We thank the reviewer for their thorough evaluation of our work and identifying the potential of our approach. Below we address the concerns raised by the reviewer:
> 1. Thank you for bringing [1] to our attention, we were not aware of it. It is an interesting method that directly modifies the kernel of the GP. We are in the process of running their method for quantitative comparison and will update our draft accordingly as soon as we have their results. Though, it seems like their method will have problems when scaling to larger problems as their covariance matrix size will increase with more negative training data as opposed to our method which handles negative training data differently.
> 2. We have added an additional set of experiments in our draft, which shows a car navigating on a road with pits and cones acting as negative constraints that are to be avoided.
>
>
> 3. * Thank you for catching this. Yes, you are correct, it should be $`p(y|\theta, \bar{X})’$. Eq(6) corresponds to the stated proposal and we have updated the text accordingly.
>     * We used the re-parameterization trick to sample from the Gaussian distribution during the inference phase. For the KL Divergence calculations in Eq(6), we directly used the analytical solutions. We have updated this detail in our draft.
>     * We initially tried minimizing the loss Eq(6) directly, but we found that the optimization was not stable as it was not balancing between the two terms in the loss functions as desired.
>     * We have added the values of $\lambda$ and $\sigma_{neg}$ in the updated draft.
> 4.  We understand the confusion regarding the random shuffling technique and have updated the text to make it clearer. In the real world navigation scenario, we will actually have positive and negative datapairs given to us (Please see Fig.1 and newly added road navigation experiment, Please refer to Section 5.2 in our updated manuscript) and it will be straightforward to define the datapairs. For the experiments shown in our paper, we assumed that we are given the set of positive datapairs consisting of the input data and the corresponding labels as positive targets. Now, since we know the positive labels corresponding to the input data, we assume that any other label assigned to the same input can be treated as a negative datapair. From this point of view, the randomly created negative datapairs are valid where we ensure no positive target is equal to negative targets for any input ‘x’.
>
> **Updates to the paper:**
> * Added new set of experiments (Section 5.2)
> * Updated [1] the related work section
> * Other bug fixes as described above.

---

### Decision · Program_Chairs · 2021-01-07
**Final Decision**

**Decision:**

Reject

**Comment:**

This paper is very pleasant to read. The reviewers also like the key idea discussed and find the targeted application interesting and practical. However, after reading the indeed interesting motivation, all four reviewers expected to see more from the evaluation section, including more challenging and realistic set-ups and clearer gains over standard methods. The reviewers also discuss how both the navigation problem as well as the GP constraint problem have been tackled in the past, often in combination (e.g. reference [1] by R1). Therefore, it would be needed to see additional experimental evaluation in line with those previous works.